# Expression of Transcription Factor ZBTB20 in the Adult Primate Neurogenic Niche under Physiological Conditions or after Ischemia

**DOI:** 10.3390/genes13091559

**Published:** 2022-08-29

**Authors:** Dimo S. Stoyanov, Martin N. Ivanov, Tetsumori Yamashima, Anton B. Tonchev

**Affiliations:** 1Department of Anatomy and Cell Biology, Faculty of Medicine, Medical University, 9000 Varna, Bulgaria; 2Research Institute, Medical University, 9000 Varna, Bulgaria; 3Department of Psychiatry and Behavioral Science, Kanazawa University Graduate School of Medical Sciences, Kanazawa 920-1192, Japan

**Keywords:** ZBTB20, non-human primate, subventricular zone, neural progenitor

## Abstract

The *Zbtb20* gene encodes for a transcription factor that plays an important role in mammalian cortical development. Recently, its expression was reported in the adult mouse subventricular zone (SVZ), a major neurogenic niche containing neural stem cells throughout life. Here, we analyzed its expression in the adult primate anterior SVZ (SVZa) and rostral migratory stream (RMS) using macaque monkeys (*M**acaca fuscata*). We report that the majority of Ki67+ cells, 71.4% in the SVZa and 85.7% in the RMS, co-label for ZBTB20. Nearly all neuroblasts, identified by their Doublecortin expression, were positive for ZBTB20 in both regions. Nearly all GFAP+ neural stem cells/astrocytes were also positive for ZBTB20. Analysis of images derived from a public database of gene expression in control/ischemic monkey SVZa, showed evidence for *ZBTB20* upregulation in postischemic monkey SVZa. Furthermore, the co-localization of ZBTB20 with Doublecortin and Ki67 was increased in the postischemic SVZa. Our results suggest that ZBTB20 expression is evolutionarily conserved in the mammalian neurogenic niche and is reactive to ischemia. This opens the possibility for further functional studies on the role of this transcription factor in neurogenesis in primates.

## 1. Introduction

The concept that neural stem cells (NSC) regenerate neurons in the adult mammalian brain is currently widely accepted [1]. The subventricular zone (SVZ) of the lateral cerebral ventricle and the subgranular zone (SGZ) of the hippocampal dentate gyrus are the two best known adult neurogenic regions (niches). Other brain regions, such as the hypothalamus, amygdala, striatum, and substantia nigra may also be able to generate new neurons throughout life [2]. The SGZ is situated adjacent to the hippocampal hillus under the dentate gyrus and is responsible for producing new dentate granule cells [1,3,4,5]. A minority of those newly formed neurons survive and become functionally integrated into the local circuits [6,7]. Hippocampal neurogenesis was also confirmed in monkeys and humans and persists throughout life [8,9,10,11]. The SVZ is located along the wall of the lateral ventricle [12,13]. It contains a heterogenous cell population including NSCs, transit-amplifying cells (TACs), and neuroblasts that are all at different levels of differentiation. These cell types are also known as B, C, and A cells, respectively. NSCs are able to form rapidly proliferating cells (TACs) that can give rise to immature neurons (neuroblasts) that migrate towards the olfactory bulb, forming a chain of migrating cells known as the rostral migratory stream (RMS) [12,13,14]. Once their final destination is reached, they mature into interneurons and functionally integrate into the local circuitry [15]. Both SVZ and RMS seem to be preserved across mammalian species but do exhibit some structural differences. The monkey RMS appears as a well-formed hypercellular stream, exhibiting similar morphological traits to the one found in rodents, but cellular migration can be scarce [16,17,18,19]. Humans also show a preserved SVZ and an RMS, but migratory cells are so far considered to be rare at adult age [20,21]. The adult primate SVZ consists of three layers. The apical-most layer, bordering the ventricular surface, is the ependymal layer (EL). Underneath it is a hypocellular zone called the gap zone, filled with astrocytic and ependymal processes originating from the adjacent layers. Deeper than the EL is a sheet of cells, many of them astrocytes, representing the subependymal layer (SEL) [17,21,22,23].

Cerebral ischemia is the leading cause of damage to the brain. Ischemia can affect the proliferation and neurogenesis of NSCs in both SVZ and SGZ [24] While most studies focused on ischemia/neurogenesis use rodent models, data from non-human primate models suggest interspecies differences [25]. A number of transcription factors have been implicated in the regulation of rodent post-stroke neurogenesis [24], suggesting that this group of molecules has a pivotal role in the postischemic response of NSCs. However, the evidence on whether and how transcription factors contribute to post-stroke neurogenesis in primates are limited.

The zinc finger transcription factor Zbtb20 is expressed in the developing nervous system where its normal levels are necessary for the development of the hippocampus, neocortex, and olfactory bulb in mice [26,27,28,29,30,31]. Recent data suggests that Zbtb20 is also expressed at different stages of adult neurogenesis, forming quiescent NSCs to neuroblasts [29]. Previous research has detected *ZBTB20* mRNA expression in the monkey SVZa [32]. This prompted us to study the expression pattern of the ZBTB20 protein in macaque monkey SVZa and RMS by means of immunofluorescence. Our results suggest that ZBTB20 is expressed in most SVZ neuroblasts and in all RMS neuroblasts, which express Doublecortin (DCX+). ZBTB20 co-stains with Ki67 in proliferating cells in both SVZ and RMS. Further, nearly all astrocytes marked by expression of the Glial Fibrillary Acidic Protein (GFAP+) were also positive for ZBTB20.

## 2. Materials and Methods

### 2.1. Experimental Animals

Tissue processing and animal handling was previously described [32]. Experiments with the monkeys were approved by the Animal Care and Ethics Committee of Kanazawa University, Japan (Approval protocols AP-031498 and AP-080920). The monkeys were kept in air-conditioned cages and had free daily access to food and water. The monkeys were 4–6 years of age at the time of the experiments. Surgical procedures for inducing brain ischemia have been previously described [32].

### 2.2. Immunofluorescence

Histological processing was performed as previously described [32]. Antigen retrieval was achieved in a Dako PT Link Pre-Treatment Module (PT100) for 5 min at 95°C in citrate buffer (pH6) and 3 PBS washes of 10 min each followed. Bovine serum diluted 1:10 in 1% of Triton X-100/PBS was applied. The primary antibodies were diluted in bovine serum/PBS/Triton. The dilutions of the primary antibodies used were as follows: ZBTB20 (1:100; HPA016815, Sigma-Aldrich, Oakville, ON, Canada, GFAP (1:800; ab4674, Abcam, Cambridge, UK), S100β (1:100; s2532 SH-B1, Sigma-Aldrich, Oakville, ON, Canada), DCX (1:500; sc-8066, Santa Cruz, Dallas, TX, USA), NeuN (1:4000; BN90, EMD, Burlington MA, USA), and Ki67 (1:50; MIB-1 M7240, DAKO, Santa Clara, CA, USA). Following 24 h of incubation with the primary antibodies, the sections were washed in PBS 3 × 5 min and incubated for 2 h at room temperature with species-directed secondary antibody conjugated to AlexaFluor-488, AlexaFluor-647, or AlexaFluor-555 fluorochromes (Thermo Fisher Scientific, Rockford, IL, USA). All secondary antibodies were diluted in bovine/PBS/Triton at 1:300. Nuclear counter staining was achieved with 4′,6-diamidino-2-phenylindole (DAPI) at 1:10,000 for 30 min. The negative control, created by omitting the primary antibody, was used to confirm reaction specificity.

### 2.3. Image Acquisition and Analysis

Motorized wide-field epifluorescence microscope Zeiss AxioImager Z.2 equipped with an AxioCam Mrm rev.3 monochrome CCD camera was used for image acquisition. Z-stacks of the entire SVZ and RMS were acquired using a 20× objective (EC Plan-Neofluar 20×/0.50 M27 at resolution of 0.322 μm/pixel) and the best focus plane was chosen for the counting. To achieve higher resolution when needed, we acquired images of target zones identified on the Z-stacks and used Apotome2.0 (ZEISS Apotome.2, White Plains, NY, USA) structured illumination and a 40× air epifluorescence objective or a 100× oil immersion objective in AxioVision SE64 Rel. 4.9.1 or ZenBlue. The obtained images were exported in .tiff format and the cells were manually counted in FiJi [33]. Statistical analysis was undertaken using R and the figure plots were produced using the package ggplot2 [34,35].

### 2.4. ISH Image Analysis

ISH images were derived from the Monkey-niche public database (http://monkey-niche.org/, accessed on 1 of February 2022) [32]. For the quantitative analysis of gene expression, we used the custom software Celldetekt (https://github.com/tumrod/cellDetekt, accessed on 2 of February 2022) [36]. This program segregates the input image into regions by binning adjacent pixels and classifies the binned regions in 4 groups based on the estimated level of expression: (1) regions filled with dye, (2) regions partially filled with dye, (3) regions with scattered puncta of dye, and (4) and regions with no dye [36]. We estimated the regions from classes 1 and 2 to represent areas with strong expression. Based on that, we calculated the presence of regions with strong expression in the area of interest. Then, we compared the present area with strong expression in control versus ischemic brains. EL and SEL were distinguished based on morphological characteristics such as relative thickness and cell location. EL was defined as a one-cell-thick layer adjacent to the cavity of the lateral ventricle, while SEL was a region subjacent to EL with 200 μm depth.

## 3. Results

### 3.1. ZBTB20 Is Expressed in the Macaque SVZa

On coronal brain sections through the adult monkey brain, we identified the caudate nucleus and the putamen separated from each other by fibers of the internal capsule, while the remaining were adjoined by the nucleus accumbens. This level corresponded to levels ac + 3 to ac + 5 (ac, anterior commissure) according to a stereological monkey brain atlas [37]. In the DAPI channel, we identified a stream of densely packed cells corresponding to the vertical limb of the RMS (Figure 1(a1)). In the GFAP channel, we identified astrocyte processes forming a glial canal which is used as the migratory route for the newly formed neuroblasts (Figure 1(c2)). On an adjacent section stained for DCX, we found DCX+ neuroblasts in the RMS (Figure 1(c3)). At the ventricular surface, ZBTB20 was strongly expressed in the ependymal layer while scattered cells were present in the SEL (Figure 1(b1)). ZBTB20 did not co-localize with mature neurons labeled by the marker NeuN in the striatum (Appendix A).

### 3.2. Immunohistochemical Characterization of ZBTB20+ Cells in Macaque SVZa

Within the EL layer, a ZBTB20 positive signal was observed in all cells with ependymal morphology, which were also positive for S100β, a calcium binding protein consistently expressed in all ependymal cells (Figure 2(c1–c4)). We next focused on the SEL which is composed of heterogeneous cell types, including parenchymal astrocytes, NSCs, TACs, and neuroblasts. GFAP is expressed by both parenchymal astrocytes and NSCs. Almost all GFAP+ cells co-expressed ZBTB20 (55 of 56 GFAP+ cells) (Figure 2(d1–d4)). We next used DCX to identify neuroblasts in the SEL and found that 84.8% (28 of 33 DCX+ cells) of them where positive for ZBTB20 (Figure 2f). We tested whether the ZBTB20+ cell proliferated by using Ki67 to visualize proliferating cells in the SVZ, and we found that 71.4% (45 of 63 Ki67+ cells) were ZBTB20+ (Figure 2e). We investigated whether ZBTB20 was expressed in TACs, defined as Ki67+/DCX- cells in SEL. We found that 68.5% (37 of 63 cells) of the Ki67+/ZBTB20+ cells were negative for DCX, and thus most probably represented TACs (Appendix A).

### 3.3. Immunohistochemical Characterization of ZBTB20+ Cells in RMS

We identified ZBTB20+ cells both within the RMS (Figure 3(d2,e2)) and in the surrounding of the RMS (Figure 3(c2)). Upon ZBTB20/GFAP co-labeling, almost all the GFAP+ cells (20 of 21 GFAP+ cells) were ZBTB20+ (Figure 3(c1–c4)). We co-stained ZBTB20 and DCX (Figure 3(d1)), which marks migrating neuroblasts within a “canal” (glial tube) of glial cells (Figure 3(c1)). Several clusters of DCX+ cells forming a “honeycomb” pattern were observed. Due to the high cell density in this region, we applied structured illumination to subtract the out-of-focus signal and thus increase resolution. With this method we estimated that nearly all RMS DCX+ cells (77 of 79 cells) were ZBTB20+ (Figure 2d). We studied for the presence and phenotype of proliferating Ki67+ cells in the monkey RMS (Figure 2(e1)). Our analyses revealed that 85.7% (114 of 133 Ki67+ cells) were ZBTB20+ (Figure 2(e1–e4)).

### 3.4. Enhanced ZBTB20 mRNA Expression in the Adult Monkey SEL following an Ischemic Insult

Ischemic brain injury is a known activator of progenitor cell proliferation and neurogenesis in the macaque SVZa [38,39]. We took advantage of the public database www.monkey-niche.org [32], which shows the in situ expression of 150 genes in the adult macaque monkey SVZa under normal and ischemic conditions. Analyzing images extracted from the database, we found a striking postischemic enhancement of the *ZBTB20* mRNA in the SEL (Figure 4). Triple-labeling for ZBTB20, Ki67, and DCX in postischemic SVZa (Figure 5) revealed that all (85 of 85 cells) of the studied Ki67+ cells were ZBTB20+ (Figure 5(b1–b4,d)). Nearly all of the studied DCX+ cells (174 of 183 cells) were ZBTB20+ (Figure 5(c1–c4,d)).

## 4. Discussion

Here, we report for the first time the phenotype of ZBTB20-expressing cells in adult primate SVZa and RMS. In rodents, Zbtb20 expression in the adult SVZ progenitor cells has been reported [29]. According to our findings, in the intact adult macaque SVZa and RMS over 70% of the Ki67+ cells and nearly all DCX+ cells and GFAP+ cells expressed ZBTB20. These phenotypic characteristics resemble the percentage of Zbtb20 co-expression with Gfap, Ki67, and Dcx in the mouse, but of note is that the level of Zbtb20 protein in Dcx+ cells in the mouse decreases as compared to levels in GFAP+ cells [29]. Outside the neurogenic region, in the adult brain, the Zbtb20 protein was not detectable in mature forebrain neurons in mice [40], and in this study we found a similar pattern in the monkey.

Altogether, our data demonstrate that ZBTB20 is expressed in diverse sets of niche cells in the adult primate SVZa niche, including NSCs, TACs, and neuroblasts, but its function remains to be identified. In the developing mouse brain, Zbtb20 loss-of-function results in a reduced NSC proliferation [28]. Ischemia is a strong promoter of progenitor proliferation in the monkey SVZa [38,39]. In the present study, we demonstrated enhanced postischemic *ZBTB20* mRNA levels in parallel with an increased percentage of ZBTB20 co-expression with Ki67 and DCX. These data suggest that *ZBTB20* is candidate regulator of primate SVZa precursor cell proliferation.

If ZBTB20 is indeed a regulator of SVZa progenitors, a number of possible mechanisms could be implicated. Research into the function of ZBTB20 on different types of cancer cells has revealed that in tumors, ZBTB20 positively regulates migration, tissue invasion, and proliferation [41,42,43]. An important mechanism for achieving its proliferative effects might come from the fact that ZBTB20 promotes the expression of the receptor for epidermal growth factor (EGFR), at least in hepatocytes [44]. In the context of adult neurogenesis, EGFR promotes the exit of NSC from quiescence and directs them toward activation and transformation into TACs, maintaining their division potential [45,46,47]. Further mechanistic experiments are required to prove a ZBTB20-EGFR link in the regulation of adult neurogenesis in primate SVZa.

## Figures and Tables

**Figure 1 genes-13-01559-f001:**
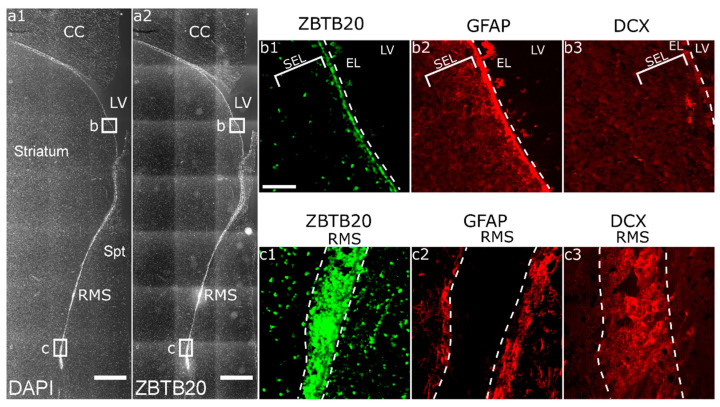
**Overview of the ZBTB20 expression in macaque SVZa.** (**a1**) DAPI staining demonstrates the major brain regions on a frontal brain section through the macaque brain. The lateral ventricle, corpus callosum, striatum, and septum are clearly identifiable. The SVZa is located on the striatal side of the lateral ventricle. RMS is well defined as a cord of densely packed cells; (**a2**) low magnification view of ZBTB20 expression on an adjacent brain section; (**b1**) higher power view of ZBTB20 expression in the SVZa; (**b2**) and (**b3**), respectively, are representative images of GFAP and DCX expression in the SVZa; (**c1**) location of ZBTB20^+^ cells in the RMS; (**c2**) the RMS is bordered by a sleeve of GFAP^+^ processes forming a glial tube; (**c3**) in the lumen of the tube, many DCX^+^ cells are present. EL—ependymal layer; CC—corpus callosum; LV—lateral ventricle; RMS—rostral migratory stream; Spt—septum. Scale bar: 1 mm (**a1**,**a2**).

**Figure 2 genes-13-01559-f002:**
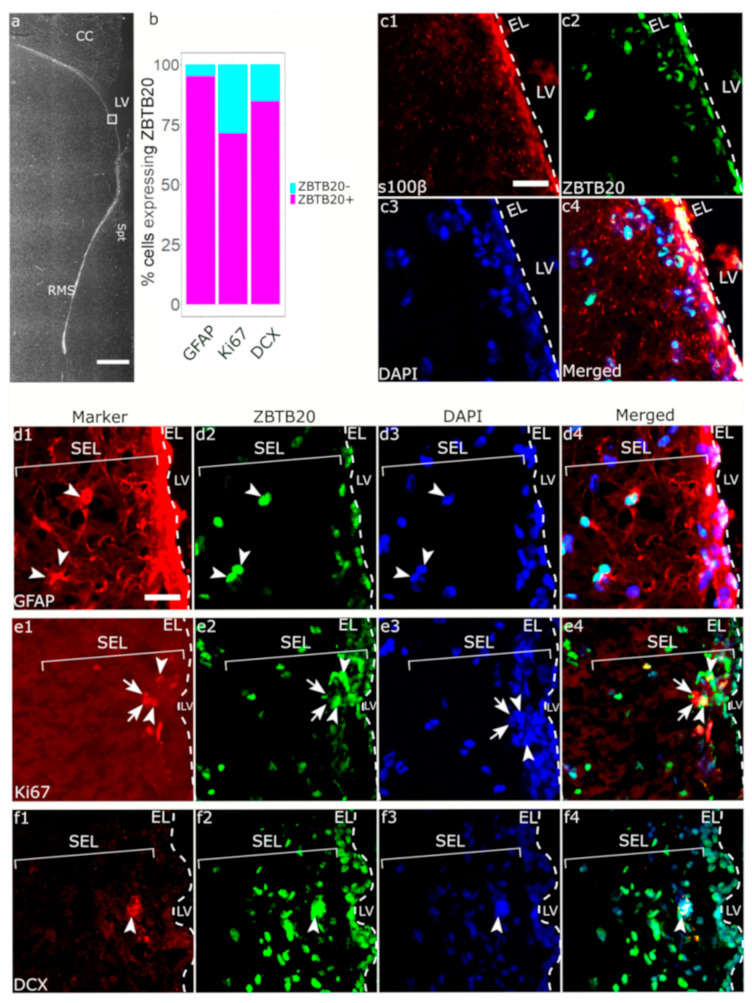
**Phenotypic characterization of ZBTB20^+^ cells in the primate SVZa.** (**a**) Low magnification image showing the relative position of the higher magnification micrographs in the other panels; (**b**) stacked bar plot summarizing the percentage of co-expression of ZBTB20 by specific cell populations defined by the markers shown on the plot; (**c1**–**c4**) the EL is defined by high s100β expression, all ependymal cells are s100β^+^/ZBTB20^+^; (**d1**–**d4**) most GFAP^+^ cells in SEL are ZBTB20^+^ (arrowheads); (**e1**–**e4**) a few Ki67^+^ cells can be detected in SEL, some are positive (arrowheads) or negative (arrows) for ZBTB20; (**f1**–**f4**) DCX^+^ cells were rare in SEL, arrows depict a DCX^+^/ZBTB20^+^ cell cluster. CC—corpus callosum; EL—ependymal layer; LV—lateral ventricle; RMS—rostral migratory stream; Spt—septum; SEL—subependymal layer. Scale bar: 1 mm (**a**), 25 μm (**c**–**f**).

**Figure 3 genes-13-01559-f003:**
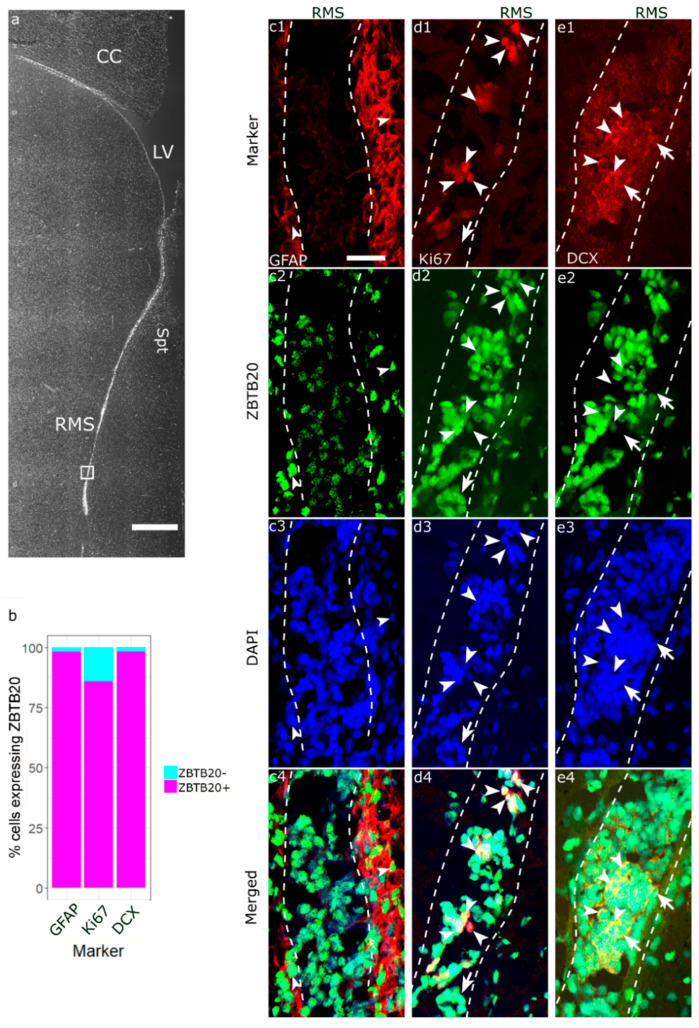
**Phenotypic characterization of ZBTB20^+^ cells in the primate RMS.** (**a**) Low magnification image showing the relative position of the higher magnification micrographs in the other panels; (**b**) stacked bar plot summarizing the percentage of co-expression of ZBTB20 by specific cell populations defined by the markers shown on the plot; (**c1**–**c4**) arrowheads depict GFAP^+^/ZBTB20^+^ cells; (**d1**–**d4**) Ki67/ZBTB20 co-staining demonstrates the presence in the RMS of numerous double-labeled cells (arrowheads) and single-labeled ZBTB20 cells (arrow); (**e1**–**e4**) DCX^+^ cells form dense clusters in the RMS exhibiting a honeycomb pattern (e1), and most co-express ZBTB20 (arrowheads). However, some DCX^+^ cells did not co-label for ZBTB20 (arrows). CC—corpus callosum; LV—lateral ventricle; RMS—rostral migratory stream; Spt—septum. Scale bar: 1 mm (**a**), 25 μm (**c**–**e**).

**Figure 4 genes-13-01559-f004:**
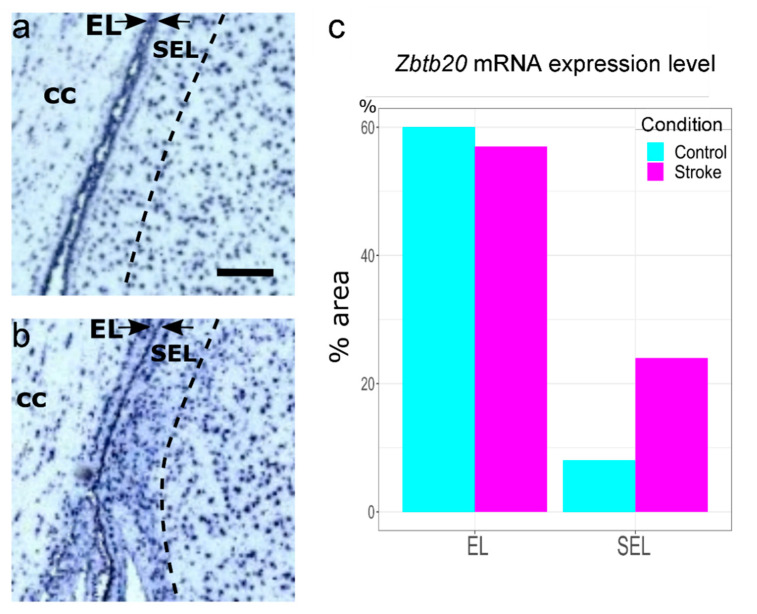
**Increased expression of *ZBTB20* mRNA in adult macaque monkey SVZa following ischemia.** (**a**) Control SVZa; (**b**) ischemic SVZa. The images were extracted from the www.monkey-niche.org public database [30]. A marked enhancement of the *ZBTB20* mRNA signal is seen in the SEL following ischemia, confirmed by quantitative assessment using the Celldetekt software (**c**). Scale = 200 μm. CC—corpus callosum.

**Figure 5 genes-13-01559-f005:**
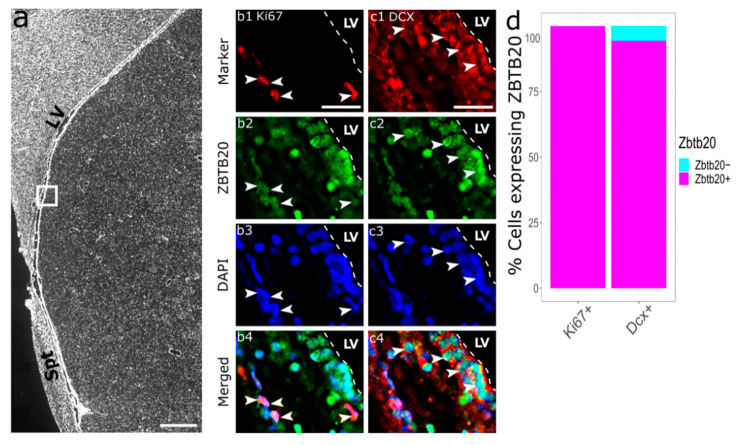
**Phenotypic characterization of ZBTB20+ cells in the postischemic primate SVZa.** (**a**) Low magnification image showing the relative position of the higher magnification micrographs panels (**b**–**c**); (**b1**–**b4**) arrowheads depict Ki67+/ZBTB20+ cells.; (**c1**–**c4**) DCX/ZBTB20 co-staining demonstrates the presence of numerous double-labeled cells (arrowheads); (**d**) stacked bar plot summarizing the percentage of ZBTB20 co-expression by specific cell populations defined by the markers shown on the plot; CC—corpus callosum; LV—lateral ventricle; Spt—septum. Scale bar: 1 mm (**a**), 25 μm (**b**,**c**).

## Data Availability

Not applicable.

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
