# Peer review of "Expression of Transcription Factor ZBTB20 in the Adult Primate Neurogenic Niche under Physiological Conditions or after Ischemia"

_genes, 2022, doi:10.3390/genes13091559_

Round 1
Reviewer 1 Report
Comments:
Stoyanov et al have shown that ZBTB20 expression is found in the neurogenic niche of adult primate model using macaque monkeys (M. fuscata). It is an interesting study where authors show co-expression of ZBTB20 with doublecortin (neuroblast marker) and other markers of neural stem cells.
Major comment-
The authors suggest that ZBTB20 mediates regulation of primate SVZa precursor cells in normal and pathological conditions like ischemia. However, this study is a correlation study and does not give any mechanistic evidence as to if and how ZBTB20 regulates of primate SVZa precursor cells.
1. It would be great if they can do an experiment showing that decreasing/increasing expression of ZBTB20 affects neural stem cells or neuroblasts.
2. The authors should show the correlation of ZBTB20 expression with DCX and Ki67 in ischemia.
Minor comment-
1. The authors should change Figure 1 (a2) Low magnification view of ZBTB20 expression on an adjacent brain section to a different image as there seems to be a repetitive band like disturbance in the image.
2. The authors should discuss about the possible pathways/ mechanisms by which ZBTB20 could be regulating the neurogenic niche.
Author Response
We thank the reviewers for the comments and submit our response.

Reviewer 2 Report
1. All of references are old. the authors should update them and use recently published paper.
2. In discussion section, the authors should compare their results with recently published paper. Therefore, they should rewrite the discussion section.
Author Response

(The authors gave the same response as above.)

Round 2
Reviewer 1 Report
The authors have satisfactorily answered the questions raised.